# Corneal Sub-Basal Nerve Plexus Regeneration Pattern following Implantable Collamer Lens Implantation for Myopia: A Prospective Longitudinal In Vivo Confocal Microscopy Study

**DOI:** 10.3390/biomedicines12030555

**Published:** 2024-03-01

**Authors:** Qiaoling Wei, Xuan Ding, Weiteng Chang, Xianjin Zhou, Rui Jiang, Xingtao Zhou, Zhiqiang Yu

**Affiliations:** 1Department of Ophthalmology, Eye and ENT Hospital, Shanghai Medical College, Fudan University, Shanghai 200031, China; dr_weiqiaoling@fudan.edu.cn (Q.W.); xuan.ding@fdeent.org (X.D.); weitten777@gmail.com (W.C.); zhouxianjin@scmc.com.cn (X.Z.); rui.jiang@fdeent.org (R.J.); 2Shanghai Key Laboratory of Visual Impairment and Restoration, Fudan University, Shanghai 200127, China; 3Ocular Trauma Center, Eye and ENT Hospital, Shanghai Medical College, Fudan University, Shanghai 200031, China

**Keywords:** corneal sub-basal nerve plexus, implantable collamer lens, corneal confocal microscopy, Visian ICL V4c, refractive surgery

## Abstract

Implantable Collamer Lens (ICL) surgery has increasingly been adopted for myopia correction in recent decades. This study, employing in vivo confocal microscopy (IVCM), aimed to assess the impact of corneal incision during ICL surgery on the corneal sub-basal nerve plexus (SNP) and adjacent immune dendritiform cells (DCs). In this longitudinal study, eyes from 53 patients undergoing ICL surgery were assessed preoperatively and postoperatively over a twelve-month period. Quantification of seven SNP parameters was performed using ACCMetrics V.2 software. Ultimately, the final analysis was restricted to one eye from each of the 37 patients who completed a minimum of three months’ postoperative follow-up. Preoperative investigations revealed a positive correlation of DC density with patient age and a negative association with corneal nerve fiber density (CNFD). Additionally, both DCs and CNFD were positively linked to spherical equivalent refraction (SER) and inversely related to axial length (AL). Intriguingly, preoperative DC density demonstrated an indirect relationship with both baseline and postoperative CNFD changes. Post-surgery, an initial surge in DC density was observed, which normalized subsequently. Meanwhile, parameters like CNFD, corneal nerve fiber length (CNFL), and corneal nerve fractal dimension (CNFrD) initially showed a decline following surgery. However, at one-year follow-up, CNFL and CNFrD displayed significant recovery, while CNFD did not return to its baseline level. This study thus delineates the regeneration pattern of SNP and alterations in DC density post-ICL surgery, highlighting that CNFD in the central cornea does not completely revert to preoperative levels within a year. Given these findings, practitioners are advised to exercise caution in older patients, those with high myopia, or elevated preoperative DCs who may undergo delayed SNP regeneration.

## 1. Introduction

Refractive surgery (RS), encompassing various techniques including advanced Implantable Collamer Lens (ICL) implantation, is recognized as an effective corrective intervention for myopia [1]. Postoperative assessments reveal that more than 90% of patients undergoing RS achieve notable improvements in uncorrected distance visual acuity (UDVA) [1,2]. Despite these favorable results, a subset of patients experiences postoperative ocular pain, frequently associated with dry eye syndrome, potentially linked to impaired recovery of the trigeminal nerve following surgery [3,4].

Unlike procedures such as laser-assisted in situ keratomileusis (LASIK), which inevitably affect the corneal sub-basal nerve plexus (SNP) throughout flap formation and stromal ablation [5,6], ICL implantation interacts differently with corneal structures, and recent findings suggest a lower incidence of postoperative dry eye in ICL patients [7,8]. However, comprehensive studies on how ICL affects the corneal SNP are limited.

In addressing this research gap, our study employs in vivo confocal microscopy (IVCM). IVCM is a noninvasive technique that provides high-resolution images of corneal cellular structures, including immune dendritiform cells (DCs) and the corneal SNP [9]. It is an important tool for identifying subtle changes in the cornea after surgery. Our study utilizes IVCM to closely examine changes in the corneal SNP after ICL implantation, an area that has not been thoroughly investigated before. To our knowledge, this is the first study which uses IVCM to specifically look at how ICL surgery affects corneal nerve morphology and DC density.

## 2. Materials and Methods

### 2.1. Study Patients

From September 2019 to July 2021, a prospective study initially enrolled 53 consecutive myopic patients who underwent ICL surgery for myopia correction at The EYE & ENT Hospital of Fudan University. The initial cohort comprised 42 females and 11 males, with an average age of 26.7 ± 6.1 years (ranging from 17 to 44 years) and a mean preoperative spherical equivalent refraction (SER) of −9.84 ± 2.90 D (ranging from −18.25 to −4.5 D). The surgeries were performed by Dr. Z. Yu. Informed consent was obtained from all participants, and the study conformed to the Declaration of Helsinki and received ethical approval from the institutional review board. Inclusion criteria were individuals over 17 years without prior corneal surgery, ocular trauma, or significant corneal opacities. Exclusion criteria encompassed a range of factors, including a history of prior vitrectomy, specific ophthalmic surgical procedures, the use of particular medications such as antidepressant drugs, suboptimal image quality obtained through IVCM, the presence of pre-existing corneal pathologies, as well as systemic diseases with the potential to impact the corneal SNP, such as diabetes mellitus, multiple sclerosis, and others.

### 2.2. Study Protocol

Participants underwent comprehensive ophthalmic examinations at baseline and at scheduled postoperative visits at day 1, week 1, month 1, and months 3, 6, and 12. Assessments included best-corrected visual acuity (BCVA), SER, intraocular pressure (IOP) measured by Canon Full Auto Tonometer TX-F, axial length (AL), and endothelial cell density (ECD) using SP-2000P and anterior chamber depth (ACD) via Pentacam HR imaging (Oculus GmbH, Wetzlar, Germany).

### 2.3. Confocal Microscopy and Image Analysis

IVCM was performed using the Heidelberg Retina Tomograph III (HRT; Heidelberg Engineering GmbH., Heidelberg, Germany) to evaluate the central corneal SNP, adhering to the established methodologies [10,11,12]. Each IVCM examination focused on the apex of the cornea by optimizing patient eye positioning and was conducted by an experienced operator (Q. Wei). Three representative images were chosen by two masked observers (W. Chang and D. Xuan) based on criteria including clarity, contrast, and absence of motion artifacts. DC density was manually calculated using Heidelberg Retina Tomograph III built-in cell count software. Image analysis was primarily conducted using the ACCMetrics software V.2 (MA Dabbah, Imaging Science and Biomedical Engineering, Manchester, UK) [13,14]. The software automatically calculated various corneal nerve parameters, including: (1) corneal nerve fiber density (CNFD)—the total number of nerves per square millimeter; (2) corneal nerve branch density (CNBD)—the density of second-order branches stemming from primary axons per square millimeter; (3) corneal nerve fiber length (CNFL)—the cumulative length of nerve fibers and their branches per square millimeter; (4) corneal nerve total branch density (CTBD)—the overall number of branches per square millimeter; (5) corneal nerve fiber area (CNFA)—the area occupied by nerve fibers per square millimeter; (6) corneal nerve fiber width (CNFW)—the average width of nerve fibers; and (7) corneal nerve fractal dimension (CNFrD)—a measure of the structural complexity of the corneal nerve network. The mean value of each parameter was determined from the three selected images.

### 2.4. Surgical Methodology

All ICL surgeries were conducted by Dr. Z. Yu, adhering to standard protocols [15]. Patients received topical antibiotics for three days before surgery, followed by dilating drops and topical anesthesia on the day of surgery. Precise corneal micro-incisions were made at the 3 o’clock temporal position (2.8 mm) and the 12 o’clock superior position (0.5 mm). The superior incision was utilized for introducing the ICL V4c, which features a central pore designed to maintain the circulation of the aqueous humor within the eye. This procedure was facilitated by the use of an injector cartridge. Sodium hyaluronate was inserted into and then removed from the anterior chamber, with viscoelastic removal confirmed via slit lamp examination. Postoperative care comprised antibiotic and steroid eye drops, prescribed four times daily for two weeks, followed by gradual tapering.

### 2.5. Statistics

Data analysis was performed using SPSS software (version 23.0, SPSS, Inc., Chicago, IL, USA), with values expressed as mean ± standard deviation. A repeated-measures ANOVA with Tukey’s HSD post hoc test was used to compare preoperative and postoperative IVCM parameters and DC density, considering *p* < 0.05 as significant. Correlations between patient age, ocular biometrics, and nerve changes were assessed using Pearson correlation analysis.

## 3. Results

In our prospective study, we initially enrolled 53 subjects. However, only those who completed postoperative assessments for a period exceeding three months were considered for the final analysis. This resulted in a subgroup of 37 patients. For those patients who had undergone bilateral ICL surgery, a single eye was chosen for inclusion in the analysis to maintain methodological consistency. This selection was performed randomly, ensuring that each of the 37 patients contributed data from only one operated eye, culminating in a dataset representative of 37 individual eyes. Follow-up visits were scheduled at 1 day, 1 week, and 1, 3, 6, and 12 months post-surgery, with attendance recorded as 36, 29, 22, 27, 12, and 12, respectively, for these time points. On average, the cohort attended 4.76 ± 1.0 follow-up sessions. The participants’ ages ranged from 17 to 41 years, with a median age of 26.6 ± 5.4 years, including 29 females. Comprehensive preoperative clinical and demographic details are presented in Table 1. It is noteworthy that all subjects maintained a post-surgical intraocular pressure below 21 mmHg. Furthermore, the arch height measurements for all participants fell within the safe range of 250–750 μm. The postoperative UDVA consistently matched or exceeded the baseline BCVA. Importantly, there were no reported instances of abnormal ICL positioning, corneal abnormalities such as edema, eye discomfort, or vision disturbances such as night vision symptoms or dry eye symptoms either before or after surgery. Additionally, no cases of ICL extraction or replacement were reported, and the study observed no serious complications, including anterior subcapsular opacity, cataract formation, secondary glaucoma, or pupillary block.

The preoperative evaluation of DC density revealed an average of 26.52 ± 50.6 cells/mm^2^, which increased to 35.07 ± 57.8 cells/mm^2^ on the first day post-surgery before generally regressing toward baseline levels in later assessments. Specific postoperative values were 21.12 ± 55.0 cells/mm^2^ at one week and 11.46 ± 10.3 cells/mm^2^ at one month, culminating in 12.98 ± 21.4 cells/mm^2^ at the one-year evaluation (as detailed in Table 2 and referred to Figure 1). Significant variations in DC density were identified through repeated-measures ANOVA and Tukey’s HSD test, particularly between the first postoperative day and the third month (Figure 2A). Pearson’s correlation analysis demonstrated a positive relationship between preoperative DC density and age (R = 0.387, *p* = 0.018) and AL (R = 0.410, *p* = 0.012), but a negative correlation was revealed with preoperative CNFD (R = −0.529, *p* = 0.001) and measurements from the third postoperative month (R = −0.422, *p* = 0.020). Furthermore, a negative association was found with SER (R = −0.461, *p* = 0.005), while a positive correlation was identified with postoperative DC density across all time points (all *p* < 0.05), with the exception of postoperative month 3. This pattern was also reflected in the preoperative CNFA (R = 0.499, *p* = 0.002) and its status on postoperative day 7 (R = 0.369, *p* = 0.045).

Postoperative analysis revealed a significant decrease in CNFD, with values declining from a baseline of 24.55 ± 9.1 nerves/mm^2^ to 17.93 ± 7.7 nerves/mm^2^ on the first day following surgery. One-year post-surgery, CNFD remained decreased at 18.22 ± 8.3 nerves/mm^2^, indicating no return to preoperative levels (Figure 3). Concurrent decreases were observed in CNFL and CNFrD, which dropped to 12.54 ± 4.2 mm/mm^2^ and 1.450 ± 0.07, respectively, compared to preoperative values of 14.71 ± 3.2 mm/mm^2^ and 1.479 ± 0.04. By the end of the first postoperative year, CNFL and CNFrD approached baseline measurements. Initial reductions were also noted in CNBD from 26.01 ± 17.5 branches/mm^2^ to 22.54 ± 17.0 branches/mm^2^ and in CTBD from 45.52 ± 27.9 branches/mm^2^ to 39.31 ± 25.5 branches/mm^2^, but by the twelfth month, values were near baseline at 26.08 ± 20.2 branches/mm^2^ and 43.55 ± 31.2 branches/mm^2^, respectively. These longitudinal changes did not reach statistical significance. However, CNFA one week post-surgery was significantly reduced (0.0049 ± 0.002 mm^2^/mm^2^) compared to preoperative values (0.0062 ± 0.002 mm^2^/mm^2^). A similar pattern was observed for CNFW, which decreased significantly to 0.0229 ± 0.002 mm/mm^2^ by the first postoperative month from a baseline of 0.0215 ± 0.002 mm/mm^2^, as detailed in Figure 2B–H.

Pearson correlation analysis conducted preoperatively identified several significant relationships with CNFD. Specifically, preoperative CNFD positively correlated with SER (R = 0.501, *p* = 0.002). Conversely, it negatively correlated with AL (R = −0.562, *p* < 0.001) and DC density (R = −0.529, *p* = 0.001). Notably, there was no significant correlation of preoperative CNFD with age, lens thickness (LT), ACD, central corneal thickness (CCT), anterior chamber volume (ACV), or ECD. Furthermore, preoperative CNFD was positively correlated with CNBD (R = 0.540, *p* = 0.001), CNFL (R = 0.748, *p* < 0.001), and CNFrD (R = 0.500, *p* = 0.002), but it was inversely related to corneal nerve fiber width (CNFW) (R = −0.476, *p* = 0.003). No significant relationships were observed between CNBD, CNFL, and CNFrD and the established ocular metrics, including age, SER, LT, AL, ACD, CCT, ACV, and ECD. In contrast, CNFA demonstrated a negative correlation with SER (R = −0.424, *p* = 0.010) and positive correlations with AL (R = 0.395, *p* = 0.015) and DC density (R = 0.499, *p* = 0.002). Additionally, CNFW showed a positive correlation with age (R = 0.384, *p* = 0.019).

## 4. Discussion

Corneal nerves are integral to corneal health, given their roles in signal transduction and neurotrophic factor secretion. RS, such as photorefractive keratectomy (PRK), LASIK, and Laser-Assisted Sub-Epithelial Keratectomy (LASEK), is known to disrupt these nerves, leading to reduced tear production and the development of ocular surface diseases [16,17,18]. IVCM has enabled the detailed observation of corneal nerves, facilitating a deeper understanding of their post-surgical responses. The impact on the SNP is highly variable and is dependent on the surgical method; thus, postoperative assessments are crucial.

The regeneration of SNPs post-PRK is notably prolonged, with full density recovery potentially taking up to two years, and morphological changes observable for up to five years [19,20]. In contrast, LASIK is associated with an acute decrease in sub-basal nerve plexus (SNP) densities, with an approximate reduction of 90% compared to pre-surgical levels. At one-year post-LASIK, less than half of the original nerve densities remain [21,22,23]. Conversely, LASEK restores corneal sensation to pre-surgical levels within three months post-procedure; however, SNP densities remain at approximately 50% of their initial values six months following surgery [24]. Moreover, regardless of the refractive technique employed—SMILE, FS-LASIK, or LASEK—patients with pronounced myopia typically do not experience a return to preoperative central SNP densities within the first-year post-surgery [25].

Our research extended the understanding of RS’s impact on corneal physiology by specifically investigating changes in immune DCs and the SNP after ICL implantation, utilizing IVCM. DCs are bone-marrow-derived and are integral to both immunological surveillance and the maintenance of corneal integrity. Increases in DC density have been observed in chronic ocular diseases, particularly those with inflammatory or infectious origins. A strong correlation has been reported between elevated DC counts and clinical signs of ocular pathology, alongside an increase in pro-inflammatory cytokines within tear fluid [26,27,28]. Following ICL surgery, our study identified a marked increase in DC density, which was confined to the immediate postoperative period, contrasting with baseline levels. This elevation in DC density was transient, with counts returning to baseline in subsequent evaluations. Several factors may account for this temporary increase. The corneal disruption induced by ICL surgery is comparatively mild, eliciting a reduced inflammatory response relative to other forms of RS. This subdued response may also be due to the minimal invasive nature of ICL surgery, which contrasts with more extensive procedures such as cataract phacoemulsification, known for longer operative times and greater disturbance of the anterior chamber. Additionally, DC distributions are more abundant around the limbus and decrease towards the central cornea [28]; thus, assessments focusing on central corneal DCs might not fully capture the scope of post-surgical inflammatory changes. Finally, the routine use of topical corticosteroids following ICL surgery is likely to suppress further DC activation and proliferation [29,30].

Consistent with the existing literature, our study demonstrated a negative correlation between DC density and CNFD both before the surgical procedure and one week postoperatively [26]. Analysis also revealed a positive correlation of DC density with patient age and axial length, whereas a negative correlation was observed with SER. The lack of a significant difference in preoperative DC densities between genders indicates that factors such as advanced age or pronounced myopia may be predictive of higher DC densities, which could in turn increase the risk of postoperative issues, including the degeneration of the SNP and heightened inflammatory responses.

In the current study, we explored the relationships between patient ocular parameters and the metrics of corneal SNP. Notably, CNFW was significantly correlated with age (R = 0.384, *p* = 0.019), while CNFD did not demonstrate a significant relationship (R = −0.133, *p* = 0.431). This finding diverges from prior research and may be reflective of the unique age distribution within our study population [31], which included 2 participants under 20 years, 25 between the ages of 20 and 9, 9 between 30 and 37, and 1 participant aged 41 years. Preoperatively, CNFD showed a positive correlation with SER (R = 0.501, *p* = 0.002) and a negative correlation with AL (R = −0.562, *p* < 0.001). Conversely, CNFA was negatively associated with SER (R = −0.424, *p* = 0.010) and positively with AL (R = 0.395, *p* = 0.015), suggesting that individuals with more severe myopia may have reduced CNFD and increased CNFA.

Following surgery, reductions were observed in both CNFD and CNFL. Twelve months postoperatively, CNFD levels had not returned to preoperative values, whereas CNFL had normalized to levels observed prior to surgery. Furthermore, CNFrD, which reflects the health and distribution of the nerve fiber network [32], decreased after surgery but returned to baseline within the first year. The degeneration of SNP observed post-ICL surgery is likely a consequence of mechanical trauma caused by the superior and temporal small corneal incisions—a nerve base [33]—complemented by subsequent inflammatory processes [34].

Comparatively, Kim J.H. et al. [35] documented a substantial decrease in sub-basal nerve density of 77%, 67%, and 79% at intervals of 1 week, 1 month, and 3 months post-cataract surgery at the temporal incision site. This reduction surpassed that observed in our study’s central corneal SNP. The divergence in findings may stem from the variances in incision location and size, which could affect the extent of mechanical damage to the corneal nerves and the resulting inflammatory response. Additional contributing factors might include demographic differences and procedural complexities; our study population was considerably younger (mean age, 26.6 ± 5.4 years) compared to that of Kim J.H. et al. [35] (mean age, 67.2 ± 8.6 years), and cataract surgery is inherently more intricate than ICL implantation. Moreover, our findings regarding postoperative CNFL did not correspond with the minor and statistically insignificant reduction in corneal sensitivity noted by Kim J.H. et al. one week after surgery, which returned to baseline after one month. This discrepancy implies that corneal sensitivity and nerve density, while related, may not directly parallel CNFL alterations, thus supporting Kim J.H. et al.’s observation of a discordance between the recovery of corneal sensitivity and the timeline of corneal reinnervation [35]. The variations highlight the potential for the functional recovery of corneal sensitivity to precede the morphological regeneration of nerves, or for alternative compensatory mechanisms, such as neural plasticity or redundant neural pathways, to mitigate the loss of nerve fibers.

The disparity between our results and those of Kim J.H. et al. [35] emphasizes that although small clear corneal incisions are utilized in both surgical approaches, the consequent effects on corneal nerve integrity and functionality can differ significantly. This underlines the imperative for a more detailed comprehension of the determinants that influence corneal nerve health following surgery, recognizing the possibility of patient-specific variability and the impact of subtle differences in surgical technique on nerve recovery and the restoration of corneal sensitivity.

Our analyses also revealed that preoperative DC density was positively correlated with postoperative densities at most intervals. An exception was noted at the third postoperative month, where an inverse relationship with CNFD was observed, suggesting that initial DC levels may predict post-surgical SNP integrity. Therefore, particular attention should be afforded to older individuals or those with severe myopia or high preoperative DC counts as these groups may be more susceptible to prolonged postoperative SNP regeneration.

Our investigation, focusing on the effect of ICL surgery on the SNP, comes with certain constraints. With a small sample size, the generalizability of the results is potentially compromised. The narrow age range of participants further constricts the extrapolation of these findings to a broader population. Additionally, a follow-up period limited to 12 months may not capture the full spectrum of long-term postoperative outcomes. The exclusion of corneal sensitivity assessments and patient subjective experience surveys from the study’s methodology also limits the depth of the impact analysis. Despite these constraints, this study provides an important preliminary exploration into the effects of ICL surgery on SNP integrity. Notably, this research identified a persisting reduction in CNFD that remained evident 12 months post-surgery. The dynamics of corneal nerve regeneration appear to be modulated by variables such as DC density, AL, and SER.

## 5. Conclusions

In summary, the findings of this study highlight the potential for long-lasting changes in corneal nerve structure following ICL surgery and emphasize the need for meticulous patient selection and postoperative care. This is particularly important for older individuals or those with pronounced myopia or elevated baseline dendritiform cell (DC) counts as these groups may be predisposed to a prolonged recovery period for the corneal sub-basal nerve plexus (SNP). While the insights obtained are valuable, they must be cautiously interpreted in light of this study’s limitations. Conducting future research with a larger, more diverse cohort and an extended follow-up period is crucial to validate these findings and to comprehensively understand the long-term effects of ICL surgery on corneal nerve structure and function.

## Figures and Tables

**Figure 1 biomedicines-12-00555-f001:**
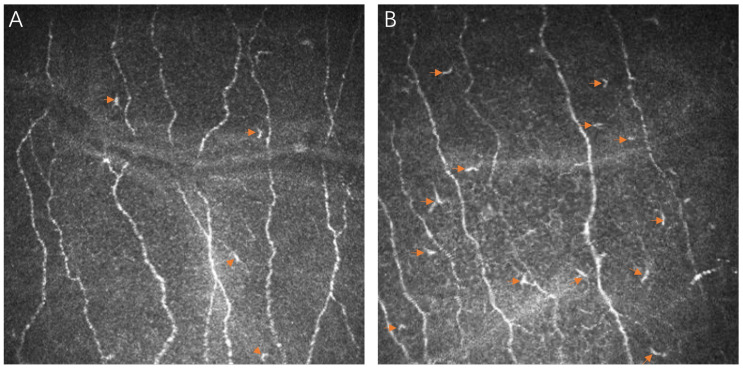
In vivo confocal microscopy images of dendritiform cells (DCs) before and one day after ICL surgery. In vivo confocal microscopy images of dendritiform cells (DCs) from the cornea of a 29-year-old female patient with high myopia, characterized by an axial length of 26.14 mm and a spherical equivalent refractive error (SER) of −7.875 diopters. (**A**) Pre-surgery: Image showing a median DC density of 25.25 ± 10.6 cells/mm^2^, with individual cells indicated by arrows. (**B**) Post-surgery (first day): Increased DC density observed, with a count of 87.5 ± 21.6 cells/mm^2^. Arrows mark the labeled DCs. Each panel represents a 400 µm × 400 µm area of the corneal layer, illustrating the changes in DC density as a response to surgical intervention.

**Figure 2 biomedicines-12-00555-f002:**
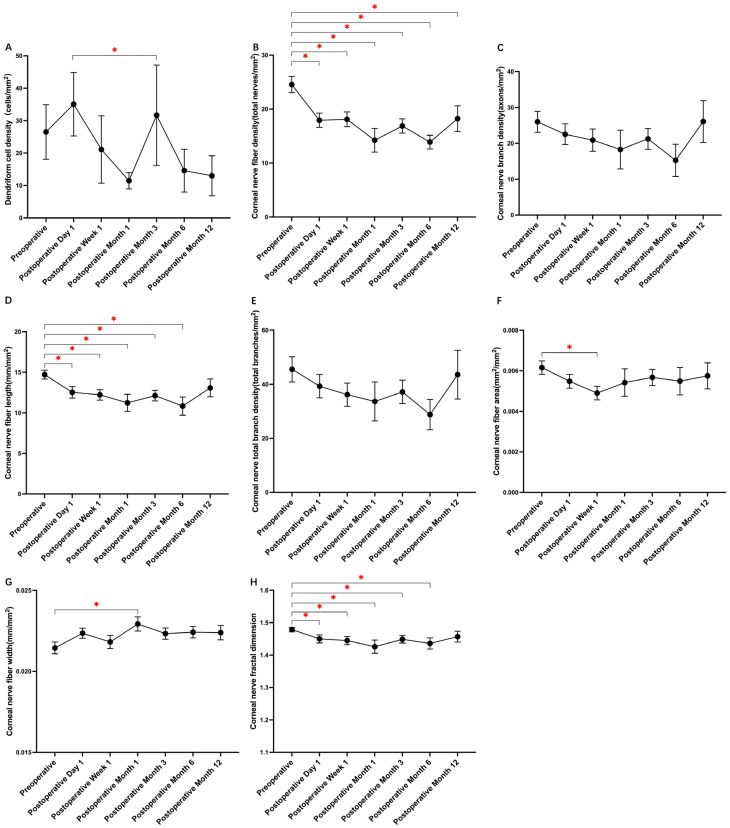
Changes in dendritic cell density and corneal nerve fibers pre- and post-ICL surgery as detected by IVCM. (**A**) Dendritic cell (DC) density: the DC density increased on the first day post-surgery compared to baseline, with the difference becoming significant only between the first postoperative day and the third postoperative month. (**B**) Corneal nerve fiber density: there was a significant decrease in density following surgery, which remained below baseline levels even at 12 months post-surgery. (**C**) Corneal nerve branch density (CNBD) and (**E**) corneal total branch density (CTBD): both metrics experienced a minor decline post-surgery but returned to baseline by 12 months post-surgery. (**D**) Corneal nerve fiber length (CNFL): a marked decrease was observed after surgery, but values returned to baseline by the 12th postoperative month. (**F**) Corneal nerve fiber area (CNFA): one week post-surgery, the CNFA values were significantly lower than baseline. (**G**) Corneal nerve fiber width (CNFW): a significant reduction was evident by the first month post-surgery. (**H**) Corneal nerve fractal dimension (CNFrD): this metric showed a notable decrease post-surgery and failed to revert to baseline levels by 12 months post-surgery. All pre- and post-surgery evaluations employed repeated-measures ANOVA, further supplemented by post hoc analyses using Tukey’s HSD test. The data depicted in the graphs are expressed as the mean ± standard error of the mean (SEM). An asterisk (*) in the figure indicates a *p*-value of less than 0.05, signifying a statistically significant difference between the compared groups.

**Figure 3 biomedicines-12-00555-f003:**
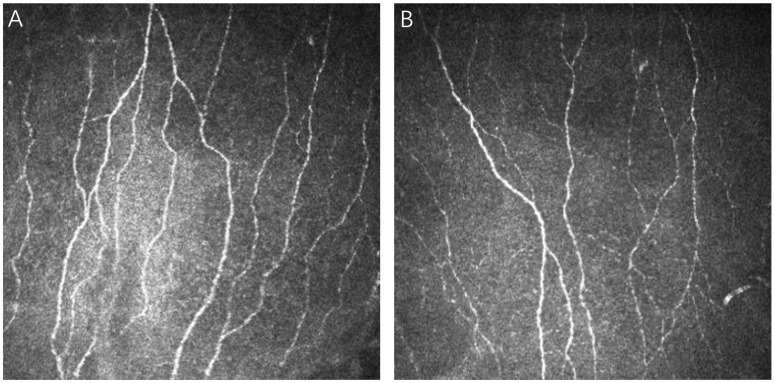
Changes in corneal nerve fiber density (CNFD) and corneal nerve fiber length (CNFL) in a 29-year-old female who underwent ICL (Implantable Collamer Lens) surgery. (**A**) Baseline (pre-surgery): prior to undergoing surgery, the CNFD and CNFL values were recorded at 37.498 ± 15.21 (total nerves/mm^2^) and 19.368 ± 11.20 (mm/mm^2^), respectively. (**B**) Postoperative month 12: by the end of the 12th month post-surgery, the CNFD and CNFL values were recorded at 24.998 ±12.11 (total nerves/mm^2^) and 14.336 ± 8.10 (mm/mm^2^), respectively.

**Table 1 biomedicines-12-00555-t001:** Overview of clinical and demographic characteristics across the entire cohort before surgery.

Parameter	Total or Mean (Range)
No. of patients	37
No. of eyes (right/left)	37 (23/14)
Gender (n, male/female)	8/29
Age (yrs)	26.6 ± 5.4 (17~41)
Spherical equivalent refraction (D)	−10.04 ± 2.70 (−16.75~−5.0)
Best-corrected visual acuity (logMAR)	0.03 ± 0.07 (−0.07~0.30)
Axial length (mm)	27.43 ± 1.4 (24.81~30.34)
Central corneal thickness (μm)	517.76 ± 29.2 (469~630)
Anterior chamber depth (mm)	3.32 ± 0.36 (2.69~4.25)
Lens thickness (mm)	3.71 ± 0.25 (3.21~4.23)
Corneal endothelial cell density (cell/mm^2^)	2998.82 ± 255.11 (2607~3653)
Anterior chamber volume (mm^3^)	200.47 ± 34.1 (136~279)

**Table 2 biomedicines-12-00555-t002:** Comparisons of corneal sub-basal nerve plexus and dendriform cell density before and after surgery.

Parameter	Before Surgery	After Surgery						*p* *
		Day 1	Week 1	Month 1	Month 3	Month 6	Month 12	
Eyes (n)	37	36	29	22	27	12	12	
DC density (number/mm^2^)	26.52 ± 50.6	35.07 ± 57.8	21.12 ± 55.0	11.46 ± 10.3	31.65 ± 84.9	14.58 ± 18.6	12.98 ± 21.4	0.008
CNFD (total nerves/mm^2^)	24.55 ± 9.1	17.93 ± 7.7	18.11 ± 7.3	14.23 ± 9.0	16.87 ± 7.1	13.89 ± 3.6	18.22 ± 8.3	0.000
CNBD (axons/mm^2^)	26.01 ± 17.5	22.54 ± 17.0	20.90 ± 16.7	18.29 ± 22.2	21.27 ± 15.6	15.27 ± 12.7	26.08 ± 20.2	0.366
CNFL (mm/mm^2^)	14.71 ± 3.2	12.54 ± 4.2	12.23 ± 3.5	11.24 ± 4.3	12.14 ± 3.6	10.84 ± 3.2	13.09 ± 3.8	0.000
CTBD (total branches/mm^2^)	45.52 ± 27.9	39.31 ± 25.5	36.18 ± 23.0	33.66 ± 29.7	37.19 ± 23.3	28.82 ± 15.8	43.55 ± 31.2	0.373
CNFA (mm^2^/mm^2^)	0.0062 ± 0.002	0.0055 ± 0.002	0.0049 ± 0.002	0.0054 ± 0.003	0.0057 ± 0.002	0.0055 ± 0.002	0.0058 ± 0.002	0.043
CNFW (mm/mm^2^)	0.0215 ± 0.002	0.0224 ± 0.002	0.0218 ± 0.002	0.0229 ± 0.002	0.0223 ± 0.002	0.0224 ± 0.001	0.0224 ± 0.002	0.045
CNFrD	1.479 ± 0.04	1.450 ± 0.07	1.445 ± 0.07	1.426 ± 0.08	1.449 ± 0.06	1.436 ± 0.05	1.457 ± 0.06	0.000

* Analysis conducted using a univariate analysis of variance (ANOVA), with statistical significance denoted as *p* < 0.05. DC, dendriform cell; CNFD, corneal nerve fiber density; CNBD, corneal nerve branch density; CNFL, corneal nerve fiber length (CNFL); CTBD, corneal nerve total branch density; CNFA, corneal nerve fiber area; CNFW, corneal nerve fiber width; CNFrD, corneal nerve fractal dimension.

## Data Availability

The data supporting the results of the current study can be found within the article.

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
