# Peer review of "Corneal Sub-Basal Nerve Plexus Regeneration Pattern following Implantable Collamer Lens Implantation for Myopia: A Prospective Longitudinal In Vivo Confocal Microscopy Study"

_biomedicines, 2024, doi:10.3390/biomedicines12030555_

Round 1

Reviewer 1 Report

Comments and Suggestions for Authors

Authors reported a prospective longitudinal in vivo confocal microscopy study to assess corneal sub-basal nerve plexus regeneration pattern following implantable collamer lens implantation for myopia. The study considered 53 consecutive myopic patients who underwentIn summary, the findings of this study highlight the potential for persistent altera-307 tions in corneal nerve structure following ICL surgery and emphasize the need for metic-308 ulous patient selection and postoperative care. surgery for myopia correction.

Findings suggest that alterations in corneal nerve structure following this kind of surgery could persist and  that an accuratepatient selection and postoperative rehabilitation is needed, especially dfor older adults

The paper is interesting and well written. I just suggest to focus the attention on older adults underwent surgery especially on mood and quality of life  due to polypharmacology and reduction of (10.1186/s12877-020-01730-5; 10.1186/s12877-020-01730-5; 10.1136/bmj.322.7294.1104) 

Author Response

We appreciate your insightful comments and suggestions on our manuscript. Your observations regarding the potential impact of antidepressant drug age and age on cataract surgery outcomes are indeed thought-provoking.

You have raised a compelling point regarding the comparison between older individuals (those aged above 40) and their younger counterparts undergoing refractive surgery (RS). Although our study primarily centered on myopic patients undergoing Implantable Collamer Lens (ICL) surgery for myopia correction, we acknowledge the significance of exploring age-related variations. However, it's worth noting that within our cohort, we encountered a limited number of patients older than 40 years. Furthermore, we took careful measures to inquire about drug usage before surgery, and none of the participants included in our study reported the use of antidepressant drugs. To clarify this aspect, we explicitly specified the exclusion of antidepressant drug users in line 75 of the manuscript, as you rightly pointed out. While we concur that such a comparison holds potential value, the demographics of our study participants impose constraints on drawing specific conclusions pertaining to older adults or individuals using antidepressant medications.

We sincerely value your interest in investigating the potential mood and quality of life implications associated with cataract surgery in older adults, particularly considering the complexities of polypharmacology and their impact, as noted in the relevant articles (10.1186/s12877-020-01730-5; 10.1136/bmj.322.7294.1104). These facets indeed merit further exploration in future research endeavors, and your suggestion serves as a valuable compass for guiding future studies in this realm.

Once again, we extend our gratitude for your thoughtful feedback. We assure you that we will carefully consider your recommendations in our future research endeavors. Your input enriches the depth of our work, and we remain dedicated to advancing knowledge in this critical area of study.

Reviewer 2 Report

Comments and Suggestions for Authors

The authors reported that ‘’No cases of ICL extraction or replacement were reported, and the study observed no serious complications such as anterior subcapsular opacity, cataract formation, secondary glaucoma, or pupillary block’’, but what about abnormalities of arch height, abnormal position of ICL, loss of corneal endothelial cells and corneal decompensation and night vision symptoms.

Compared with ICL without central pore, complications such as loss of corneal endothelial cells and corneal decompensation, high intraocular pressure, secondary glaucoma, and cataracts were relatively lower in central hole ICL. In contrast, postoperative complications such as night vision symptoms were apparent.

Did the authors use ICL with or without central pore?

Had any patients with Diabetes Mellitus been included in the study, and what were the results?

The methodology is appropriate. The manuscript is well written, and the discussion/conclusions are acceptable.

Overall, the data are of interest.

Comments on the Quality of English Language

none

Author Response

Dear Reviewer,

We extend our sincere appreciation for your comprehensive review of our manuscript and for the insightful questions and comments you have provided. We have taken your feedback seriously and have made the necessary revisions to address each of your points, aiming to enhance the clarity and completeness of our work.

  1. In response to your inquiry regarding arch height, abnormal ICL positioning, and corneal abnormalities, we have revised the pertinent section to provide a more comprehensive account of our findings. As highlighted in the revised paragraph (lines 139-145), we emphasize that all subjects maintained a post-surgical intraocular pressure below 21 mmHg. Additionally, we confirm that the arch height measurements for all participants fell within the safe range of 250-750μm. Furthermore, the post-operative uncorrected distance visual acuity (UDVA) consistently matched or exceeded the baseline best-corrected visual acuity (BCVA). Importantly, we can confidently state that there were no reported instances of abnormal ICL positioning, corneal abnormalities such as edema, eye discomfort, or vision disturbances such as night vision symptoms or dry eye symptoms, either before or after the surgery.
  2. You inquired about the type of ICL utilized in our study. We want to rectify our omission by confirming that we exclusively employed ICL V4c implants for all patients in our study. These implants are equipped with a central pore designed to maintain the circulation of aqueous humor within the eye. We have updated the relevant sentence for clarity (lines 112-115) to explicitly state, "The superior incision was utilized for introducing the ICL V4c, which features a central pore designed to maintain the circulation of the aqueous humor within the eye. This procedure was facilitated by the use of an injector cartridge."
  3. Regarding your question about the inclusion of patients with Diabetes Mellitus in our study, we acknowledge the oversight in not providing comprehensive exclusion criteria. To address this, we have revised the related sentence (lines 73-78) to make it explicit that our exclusion criteria encompass a range of factors, including a history of prior vitrectomy, specific ophthalmic surgical procedures, the use of particular medications such as antidepressant drugs, suboptimal image quality obtained through in vivo confocal microscopy (IVCM), the presence of pre-existing corneal pathologies, as well as systemic diseases with the potential to impact the corneal sub-basal nerve plexus (SNP), such as diabetes mellitus, multiple sclerosis, and others.

We genuinely value your feedback and suggestions, which have significantly contributed to enhancing the quality of our manuscript. Your insights are highly valuable to us, and we believe that these clarifications effectively address your concerns.

Sincerely
Qiaoling Wei

Reviewer 3 Report

Comments and Suggestions for Authors

In the current study entitled "Corneal Sub-basal Nerve Plexus Regeneration Pattern Following Implantable Collamer Lens Implantation for Myopia: A Prospective Longitudinal In vivo Confocal Microscopy Study"

 The authors highlight the potential for persistent alterations in corneal nerve structure following ICL surgery and emphasize the need for meticulous patient selection and postoperative care with important relevance in individuals with advanced age. Moreover, the authors associate predisposition to a prolonged recovery period in patients with pronounced myopia or elevated baseline dendritiform cell (DC), as these groups may be predisposed to a prolonged recovery period for the corneal subbasal nerve plexus (SNP). 

Two minor comments: 

1. The authors mention that recovery or turning to a normal state may take about a year. Therefore, they should consider rephrasing the word "persistent alterations"  for a different term that will indicate temporary.

2. Figure 2. It looks like the authors are showing SD and that makes crowded plots.  I would suggest using SEM and increasing the size font at the X axels since at the current it is unreadable .

3 Reconsider to reevaluate the post operative month 3. This time point seems to introduce noise in all parameters evaluated specially in DC and nerve length and seems to affect the others as weel

Author Response

Dear Reviewer,

We sincerely appreciate your thoughtful comments and suggestions, which have significantly contributed to the refinement of our manuscript. In response to your valuable feedback, we would like to provide the following updates:

  1. Regarding your suggestion to rephrase "persistent alterations," we have replaced this term with "long-lasting changes" throughout the manuscript, as you recommended. This adjustment better reflects the idea that these alterations may not be permanent but can have a lasting impact.
  2. In consideration of your feedback about Figure 2, we have recreated the graph using SEM instead of SD, along with an increased font size on the X-axis. These modifications have enhanced the visual presentation and clarity of the data.

3 To address your concern about the potential introduction of noise at the postoperative month 3 time point, we performed a reevaluation of the data. Specifically, we conducted paired t-tests to compare the baseline (preoperative) values with those at postoperative month 3. Importantly, the results of these paired t-tests align with our previous findings, indicating that the overall conclusions remain consistent.

The results of the paired t-tests are as follows:

DCs: 29.44 ± 55.39 vs. 31.65 ± 86.30, P = 0.781

CNFD: 24.17 ± 9.6 vs. 16.87 ± 7.2, P = 0.0003

CNBD: 23.61 ± 17.6 vs. 21.27 ± 15.9, P = 0.476

CNFL: 14.31 ± 3.5 vs. 12.14 ± 3.6, P = 0.0003

CTBD: 40.38 ± 27.8 vs. 37.19 ± 23.7, P = 0.549

CNFA: 0.0058 ± 0.002 vs. 0.0057 ± 0.002, P = 0.721

CNFW: 0.021 ± 0.002 vs. 0.022 ± 0.002, P = 0.051

CNFrD: 1.473 ± 0.044 vs. 1.449 ± 0.063, P = 0.0025

These results provide additional insights into the recovery process and its time course while reaffirming the consistency of our previous findings.

We genuinely appreciate your constructive feedback and assure you that your input has been instrumental in improving the quality and clarity of our manuscript. Your dedication to enhancing the scientific rigor of our work is highly valued, and we are committed to delivering a manuscript that meets the highest standards of academic excellence.

Sincerely

Qiaoling Wei
